# RANKL Is Independently Associated with Increased Risks of Non-Alcoholic Fatty Liver Disease in Chinese Women with PCOS: A Cross-Sectional Study

**DOI:** 10.3390/jcm12020451

**Published:** 2023-01-05

**Authors:** Nan Lu, Chang Shan, Jia-Rong Fu, Yi Zhang, Yu-Ying Wang, Yu-Chen Zhu, Jie Yu, Jie Cai, Sheng-Xian Li, Tao Tao, Wei Liu

**Affiliations:** Department of Endocrinology and Metabolism, Renji Hospital, School of Medicine, Shanghai Jiaotong University, Shanghai 200127, China

**Keywords:** receptor activator of NF-κB ligand, non-alcoholic fatty liver disease, polycystic ovary syndrome, free androgen index, insulin resistance

## Abstract

Women with polycystic ovarian syndrome (PCOS) are more likely to have non-alcoholic fatty liver disease (NAFLD) than non-PCOS women; however, the exact mechanism underlying this trend is unknown. The receptor activator of NF-κB ligand (RANKL) is strongly involved in bone metabolism and has multiple functions. Recent studies suggest that RANKL is implicated in hepatic insulin resistance (IR), which is the highest risk factor for NAFLD. This study aimed to assess the role of RANKL in NAFLD in Chinese women with PCOS. A cross-sectional observational study was conducted on women newly diagnosed with PCOS, which included 146 patients with NAFLD and 142 patients without NAFLD. Sex hormones, glucose, insulin, and lipids were measured, and anthropometric data were collected. The concentration of serum total RANKL was measured using commercial ELISA kits. PCOS patients with NAFLD had a significantly higher mean age, body mass index (BMI), waist circumference (WC), and worsened metabolic profile than non-NAFLD subjects. The concentrations of high-sensitivity C-reactive protein, total cholesterol, and low-density lipoprotein cholesterol increased with the RANKL tertile (*p* for trend = 0.023, 0.026, and 0.035, respectively). A significantly positive association was found between RANKL (per SD change) and the risks of NAFLD (OR = 1.545, 95% CI = 1.086–2.199) after adjusting for confounders, including demographic factors, metabolic markers, and sex hormones. Subgroup multivariate logistic analyses stratified by age, BMI, and WC showed the same tendency. In addition, the positive association between RANKL and NAFLD seemed more prominent in lean patients with a BMI < 24 kg/m^2^ (OR = 1.70, 95% CI = 1.06–2.75) when compared to overweight/obesity subjects. Therefore, this study suggests that RANKL is positively associated with the increased risk of NAFLD in Chinese women with PCOS, independent of metabolic and reproductive factors.

## 1. Introduction

Polycystic ovary syndrome (PCOS), as a common metabolic disorder, affects 5% to 10% of woman of childbearing age worldwide [1]. Various symptoms of this multifactorial disorder occur during the reproductive years, including hyperandrogenemia, oligomenorrhea or amenorrhea, and ultrasound findings of polycystic ovaries [2]. Moreover, PCOS is associated with an increased risk of multiple metabolic diseases, including hypertension, dyslipidemia, diabetes, cardiovascular diseases [3], and non-alcoholic fatty liver disease (NAFLD) [4]. Numerous studies have revealed that insulin resistance (IR) substantially contributes to the onset and progression of PCOS and that visceral adiposity and chronic inflammation may also play a role [5]. These pathophysiological features are also shared with NAFLD.

NAFLD is characterized by abnormal fat deposition in hepatocytes regardless of alcohol abuse, which encompasses a broad spectrum of conditions from simple steatosis and non-alcoholic steatohepatitis (NASH) to cirrhosis or hepatocellular carcinoma [6]. Our previous study and other investigations suggested that the incidence of NAFLD was significantly increased in women with PCOS, regardless of obesity [7,8]; a biopsy study showed that PCOS was independently related to severe NASH and advanced liver fibrosis [9]. In addition, NAFLD was significantly more prevalent in non-obese women with PCOS than those without PCOS [10]. It has been well-established that NAFLD is a hepatic manifestation of metabolic syndrome and has been increasingly viewed as a precursor of Type 2 diabetes [11,12]. The coexistence of NAFLD will further aggregate metabolic disturbances in PCOS reversely. Thus, it is an important subgroup that needs special attention. Although current research suggests that IR and hyperandrogenemia are vital contributors [7,13], the mechanisms underlying NAFLD in PCOS patients are still incompletely understood.

Recent discoveries found that bone serves as not only a scaffold organ but also an endocrine organ, regulating energy metabolism through different osteokines [14,15]. Interactions between the hepatic and skeletal systems have gradually been disclosed [16]. The receptor activator of the NF-κB ligand (RANKL), as a tumor necrosis factor (TNF) superfamily member, is crucial for bone remodeling. Moreover, RANKL affects immunity, atherosclerosis, glucose metabolism, chronic inflammation, and skeletal muscle function [17,18]. RANK, RANKL’s cognate receptor, has been confirmed to be expressed in the human liver [19]. RANKL signaling has been shown to participate in hepatic IR in a mouse model [20]. In addition, a case study reported that the liver enzymes of a biopsy-confirmed NASH patient improved significantly during the treatment of osteoporosis with the RANKL-specific antibody denosumab [21]. 

The above results suggest a potential link between RANKL and NAFLD pathogenesis. At present, few studies have focused on the relationship between RANKL and PCOS, and the consequences of these studies are inconsistent [22,23,24]. More importantly, studies have yet to focus on the correlation between RANKL and the risk of NAFLD in women with PCOS. Therefore, the current cross-sectional study assessed the role of RANKL in NAFLD among Chinese women with PCOS.

## 2. Materials and Methods

### 2.1. Participants

We consecutively recruited PCOS patients admitted to the Endocrinology outpatient clinics of Renji Hospital from July 2015 to June 2018. The diagnosis of PCOS is based on the 2003 Rotterdam criteria: (1) biochemical and/or clinical signs of hyperandrogenemia; (2) amenorrhea, anovulation, or oligomenorrhea; (3) ultrasound findings of polycystic ovaries. Before PCOS diagnosis, the following causes of hyperandrogenemia and ovulatory dysfunction were excluded: androgen-producing tumors, Cushing’s syndrome, 21-hydroxylase-deficient non-classic adrenal hyperplasia, thyroid dysfunction, or hyperprolactinemia [8]. One experienced gastroenterologist performed the abdominal ultrasound, and the diagnosis of NAFLD followed the 2012 American Gastroenterological Association criteria [25]. Patients with viral hepatitis, alcohol abuse, and drug-induced, genetic, and autoimmune diseases were excluded. All patients were newly diagnosed with PCOS and did not take any medications that affected sex hormone levels and bone metabolism (e.g., oral contraceptives, anti-androgens, insulin sensitizers, glucocorticoids, bisphosphonates, and supplemental vitamin D). Patients with diseases that affect bone metabolism, including hyperthyroidism and hyperparathyroidism, Paget’s disease, and rheumatoid arthritis, were also excluded. A total of 288 PCOS patients aged 16–38 years, including 146 with NAFLD and 142 without NAFLD, were enrolled in the cross-sectional study (Figure 1).

The current study, approved by the Research Ethics Committee of Renji Hospital, was performed following the Declaration of Helsinki. Written, informed consent was obtained from all study subjects.

### 2.2. Data Collection and Biochemical Measurements

Biochemical assays were described previously [8]. Briefly, during the early follicular phase, peripheral venous blood was collected for laboratory analysis after 12 h of fasting. If the patient presented with amenorrhea for more than three months, fasting blood samples were collected during a bleeding episode after progestin withdrawal. All participants underwent standardized clinical evaluation, including the quantification of low-density lipoprotein cholesterol (LDL-c), high-density lipoprotein cholesterol (HDL-c), triglycerides (TG), total cholesterol (Tch), uric acid (UA), fasting blood glucose (FBG), and alanine aminotransferases (ALT) by Roche/Hitachi analyzers using Roche reagents (D 2400 and E 170 Modular Analytics modules; Roche Diagnostics, Indianapolis, IN, USA). Levels of fasting serum insulin (FINS), luteinizing hormone (LH), follicle-stimulating hormone (FSH), estradiol (E2), total testosterone (TT), dehydroepiandrosterone sulfate (DHEAs), sex hormone-binding protein (SHBG), prolactin (PRL), and androstenedione (A2) were measured using chemiluminescence (Elecsys Auto analyzer, Roche Diagnostics). High-sensitivity C-reactive protein (hs-CRP) was measured using immunonephelometric methods (Dade Behring, Deerfield, Germany). A homeostasis model assessment of insulin resistance (HOMA-IR) was calculated using the following formula:$\;\mathrm{HOMA}-\mathrm{IR}\;=\left(\mathrm{fasting}\;\mathrm{insulin}\;\left[\mathrm{mIU}/L\right]\times \;\mathrm{fasting}\;\mathrm{glucose}\;\left[\mathrm{mmol}/L\right]\right)/22.5)$. The free androgen index (FAI) was calculated using the following formula: $\mathrm{FAI}\;=\left(\mathrm{TT}\;\left[\mathrm{nmol}/L\right]\times 100\right)/\mathrm{SHBG}\;\left[\mathrm{nmol}/L\right]$. BMI was calculated using the following formula: $\mathrm{BMI}\;=\;\mathrm{weight}\;\left(\mathrm{kg}\right)/\mathrm{height}\;\left(m^{2}\right).\;$BMI ≥ 24 kg/m^2^ was defined as obese/overweight, and WC ≥ 80 cm was defined as abdominal obesity in females based on Chinese criteria [26]. 

The concentration of serum total RANKL was determined in duplicate by using a commercial ELISA kit (DY626, R&D Systems, Minneapolis, MN, USA) following the instructions from the manufacturer and previous research [27]. Total RANKL measures free soluble RANKL and osteoprotegerin (OPG)-bound RANKL, thus reflecting the level of circulatory RANKL better. The serum samples were appropriately stored at −80 °C until analysis. A standard curve was used in each assay, and samples were diluted to the linear range of the curve. Absorption at 450 nm was determined by a microplate reader. The intra- and inter-assay coefficients of variation were both less than 10%.

### 2.3. Statistical Analysis 

Statistical analyses were conducted by using SPSS software (version 20.0) (SPSS Inc., Chicago, IL, USA) and EmpowerStats software (www.empowerstats.com, accessed on 10 July 2022, X&Y solutions, Inc., Boston, MA, USA). Data are presented as median and interquartile ranges, and comparisons between groups were performed by a non-parametric Wilcoxon rank-sum test. The trend analysis of clinical features between RANKL tertiles was performed through linear regression analysis. The independent effect of RANKL (per SD change) on the risk of NAFLD was assessed by multivariable logistic regression after adjusting for potential confounders. Potential confounders were included based on clinical significance, and the number of covariates met the commonly used 10 EPV (10 events per variable) criterion for reliable analysis. The variance inflation factor (VIF > 10) was also considered to avoid collinearity. Finally, 14 variables were included in the fully adjusted model. Model 1 adjusted for demographic confounders, including age, BMI, and WC. Model 2 additionally accounted for metabolic traits, including hs-CRP, UA, TG, HDL-c, LDL-c, and HOMA-IR. Model 3 further adjusted for sex hormones, including LH, FSH, E2, and FAI. Stratified analyses were conducted according to BMI (<24 and ≥24 kg/m^2^), WC (<80 and ≥80 cm), and average age (<25 and ≥25 years). The *p*-values, 95% confidence intervals (CI), and estimated odds ratios (ORs) were reported. A *p*-value smaller than 0.05 was considered as statistically significant.

## 3. Results

### 3.1. Biochemical and Clinical Features of PCOS Patients Stratified by the Presence of NAFLD

The background demographic and biochemical characteristics of PCOS patients, stratified by the presence of NAFLD, are summarized in Table 1. The mean age, BMI, WC, and concentrations of hs-CRP and ALT were higher in PCOS patients with NAFLD than those without NAFLD. Furthermore, the NAFLD group showed a worsened metabolic profile than patients without NAFLD. UA, TG, Tch, LDL-c, FBG, FINS, and HOMA-IR levels were higher and HDL-c was lower in the NAFLD group. As for the sex hormone levels, LH, FSH, and SHBG were significantly lower and FAI was more elevated in the NAFLD patients. There were no significant differences between the groups in the levels of PRL, E2, TT, DHEAs, and A2. The difference in RANKL between the two groups did not reach significance (34.0 pg/mL vs. 40.3 pg/mL, *p* = 0.337), probably accounting for the unmatched demographics (age, BMI, and WC) between groups. Thus, further multivariate analysis was needed.

### 3.2. Characteristics of PCOS Patients Stratified by RANKL Tertiles

The clinical features of the participants classified based on RANKL tertiles are shown in Table 2. The concentrations of hs-CRP, Tch, and LDL-c increased as the RANKL tertile increased (*p* for trend = 0.023, 0.026, and 0.035, respectively). The RANKL tertile increased as age decreased, although with a marginal significance (*p* for trend = 0.051).

### 3.3. Association between NAFLD and RANKL

The effect of RANKL on NAFLD in PCOS patients was analyzed through multivariate logistic regression (Table 3). After adjusting for potential confounders, including demographic factors (age, BMI, WC), chronic inflammation and metabolic markers (hs-CRP, UA, TG, HDL-c, LDL-c, HOMA-IR), and sex hormones (LH, FSH, E2, and FAI), a significant and independent positive association was identified between RANKL and NAFLD. In the fully adjusted Model 3, the OR of NAFLD per one SD increase in RANKL was 1.545 (95% CI = 1.086–2.199).

### 3.4. Subgroup Analyses

To confirm the robustness of the results, we performed a sensitivity analysis by stratifying confounders (Figure 2). All analyses were adjusted for age, BMI, WC, hs-CRP, HOMA-IR, UA, TG, HDL-c, LDL-c, LH, FSH, E2, and FAI. There was a consistent pattern across subgroups, regardless of which variable was stratified. We found that higher RANKL levels were associated with higher risks of NAFLD, with ORs varying from 1.51 to 1.96 (Figure 2).

Notably, the stratified multiple logistic regression analysis results in Table 4 suggested a more prominent association between RANKL and NAFLD in lean PCOS patients than in overweight/obesity patients. Only in PCOS patients with a BMI < 24 kg/m^2^ did the positive relationship between higher RANKL and increased NAFLD risks remain statistically significant after correcting for different regression models (Model 1, OR = 1.45, *p* = 0.04; Model 2, OR = 1.61, *p* = 0.02; Model 3, OR = 1.70, *p* = 0.03).

## 4. Discussion

To our knowledge, this study is the first to report an independent association between RANKL and NAFLD in Chinese women with PCOS. After fully adjusting for demographic factors, chronic inflammation markers, metabolic traits, and sex hormones, a higher RANKL level was independently associated with increased risks of NAFLD. This association remained consistent in subgroup analyses based on age, BMI, and WC. The results of the subgroup analysis also suggested that the positive association between RANKL and NAFLD is prone to be more prominent in lean PCOS patients with a BMI < 24 kg/m^2^.

Mounting evidence has indicated the increased prevalence of NAFLD in young women with PCOS, regardless of BMI or the presence of metabolic syndrome [28]. A recent meta-analysis including 17 observational studies involving 2561 age- and BMI-matched controls and 2734 women with PCOS demonstrated a more than two-fold higher risk of NAFLD in PCOS patients [29]. Another large population-based study identified PCOS as a risk factor for more severe histologic features of NAFLD despite being seven years younger than women without PCOS [30]. Furthermore, having NAFLD during pregnancy increases risks for gestational diabetes mellitus (GDM) [31], hypertensive complications, postpartum hemorrhage, and preterm birth [32]. Therefore, PCOS patients with NAFLD deserve particular attention. Previous studies have documented that IR, excess weight, hyperlipidemia, and chronic inflammation predispose women to NAFLD development in PCOS [33]. Hyperandrogenism may also be implicated in NAFLD through direct action in hepatic lipid metabolism [7]. However, in this study, we identified an independent association between RANKL and the increased risks of NAFLD after adjusting for age, BMI, WC, chronic inflammation marker hs-CRP, and metabolic traits, including UA, TG, HDL-c, LDL-c, and HOMA-IR. Significance persisted even after further adjustment for sex hormone factors, including LH, FSH, E2, and FAI. 

RANKL, also known as tumor necrosis factor ligand superfamily member 11 (Tnfsf11), secreted predominantly by osteoblasts, has been identified as a fundamental regulator of bone remodeling and an important factor connecting bone and energy metabolism [18,34]. Consistent with our findings, several studies have confirmed the relationship between RANKL and other metabolic diseases. A recent study reported statistically significant differences in RANKL gene methylation between obese subjects and controls, indicating the potential roles of RANKL in the pathogenesis of obesity [35]. In terms of glucose metabolism, a prospective clinical study found that serum RANKL concentration was an independent risk factor for Type 2 diabetes (T2DM) [20]. Another clinical study demonstrated that glycemic control improved in prediabetic and T2DM patients with osteoporosis who were treated with the RANKL-specific antibody denosumab, compared with those treated with bisphosphonates or calcium plus vitamin D, and the effect persisted after adjusting for BMI [36]. Meanwhile, an elevated RANKL level has been reported as an important predictor of cardiovascular disease [37,38].

Some animal studies suggested a possible relationship between RANKL and NAFLD. In a high-fat-diet-induced rat NASH model, the levels of RANKL were increased [39]. In another animal study, the progression from hepatic steatosis to NASH was accompanied by increased circulating RANKL levels [40]. However, there is little clinical evidence regarding the association between RANKL and NAFLD, and the results are not unambiguous. Some research that discussed the relationship between RANKL’s inhibitor OPG and NAFLD have provided clinical hints for the possible role of RANKL in NAFLD [41,42,43]. OPG is secreted by osteoblasts and inhibits RANKL’s function as its decoy receptor. Recently, a case–control study in T2DM showed that the OPG level was negatively associated with NAFLD independently of potential confounders [42]. However, another study in post-menopausal women with T2DM found that circulating RANKL levels were similar in patients with and without steatosis but lower in patients with NASH [44]. Additionally, in an illuminating case report, denosumab, a RANKL-specific antibody for osteoporosis treatment, unexpectedly improved liver injury in an osteoporotic patient who had NASH simultaneously [21]. Our study provided new evidence for the clinical association between RANKL and NAFLD.

There are several experimental pieces of evidence that may provide explanations of the possible mechanism. Firstly, chronic inflammation, which often involves the activation of the NF-kB pathway, is strongly implicated in insulin resistance and consequent NAFLD development [6]. As a typical activator of the NF-kB pathway, it is reasonable for RANKL to play a potential role in NAFLD, partially by the inflammatory pathway. Consistent with this hypothesis, as shown in our study, RANKL is positively associated with hs-CRP levels in PCOS patients (*p* for trend = 0.023, Table 2). Secondly, Kiechl et al. have demonstrated that RANKL regulates hepatic insulin sensitivity, and blocking RANKL signaling enhances insulin sensitivity [20]. Moreover, recent studies suggest that hepatic macrophages are involved in NAFLD progression by promoting hepatic fat accumulation [45]. Another study demonstrated that RANKL participated in Runx2-induced macrophage migration in mice with NAFLD [40]. These experimental findings and our results suggest that RANKL might serve as a therapeutic target for NAFLD. Nonetheless, larger-scale clinical studies are necessary to confirm these findings further.

Our findings also imply a subgroup in PCOS who are prone to be more susceptible to RANKL in terms of the risk of NAFLD. The positive correlation between RANKL and NAFLD remained statistically significant in lean PCOS patients after adjusting for potential confounders (Figure 2 and Table 4). This unexpected result warrants further discussion. The prevalence of non-obese NAFLD is increasing worldwide [46,47], and studies have found that non-obese NAFLD is associated with even higher all-cause mortality than obese NAFLD [48]. Notably, women with PCOS were more likely to develop non-obese NAFLD than those without PCOS [10]. In addition, the diagnosis of PCOS at a younger age was associated with more severe NASH in a homogenous population, independent of BMI [9]. These findings indicate the need to further explore the potential role of RANKL in non-obese NAFLD in PCOS women and its possible mechanisms.

Although insulin resistance and central obesity are the major risk factors for non-obese NAFLD, genetic factors, including heritability [48] and ethnicity [49] play an essential role in disease pathogenesis. SREBP2, the nuclear transcription factor that serves as the master regulator of cholesterol metabolism, is deemed as an important candidate for genetic susceptibility of non-obese NAFLD. A 7 year follow-up prospective cohort study identified the function of an SREBP2 polymorphism (rs133291) in non-obese NAFLD and its association with a higher OR for NASH (odds ratio = 2.92, 95% CI = 2.08–4.18, *p* = 0.002) [50]. Mechanistically, the activation of SREBP2 increases cholesterol accumulation in hepatocytes by coordinating cholesterol biosynthesis, uptake, and secretion [40]. Laboratory and human studies indicated that changes in cholesterol metabolism might be associated with non-obese NAFLD development and liver injury severity. For instance, excess cholesterol intake caused NAFLD in mice even without weight gain [51]. Furthermore, the oxidized LDL antibodies/HDL-c ratio is associated with advanced NAFLD in patients with a normal weight [52]. 

The above results suggest that SREBP-2-mediated abnormal cholesterol metabolism has an important role in non-obese NAFLD. In our cohort, Tch and LDL-c concentrations were higher in the upper RANKL tertile (Table 2). Given that SREBP2 and RANKL are both involved in disorders of cholesterol metabolism in non-obese subjects with NAFLD, we boldly speculate that SREBP2 might mediate the role of RANKL in NAFLD in young PCOS women with normal weight. The latest literature may provide evidence to support this hypothesis. First, a genome-wide analysis identified SREBP2 as a novel transcription factor involved in RANKL-induced osteoclast differentiation [53]. Second, SREBP2 expression was activated by the RANKL/cAMP-response element-binding protein signaling pathway [54] and was mediated by RANKL-induced reactive oxygen species [55] in osteoclast differentiation. Third, SREBP2 inhibitors inhibit RANKL-induced osteoclast differentiation [56,57,58]. Overall, the above findings demonstrate a close interaction between SREBP2 and RANKL, at least in osteoclasts, which may also occur in non-obese NAFLD. Therefore, follow-up studies on the interaction between RANKL and SREBP2 in hepatocytes might provide additional insights into the mechanisms of NAFLD. 

In the skeletal system, estrogen is one of the important regulators of RANKL, whereas few studies have examined the potential effect of RANKL on female fertility. However, as a member of TNF superfamily, RANKL plays multiple roles through the activation of the NF-kB signaling pathway and has been confirmed to be a vital player in immunity [19], mammary gland development during pregnancy [59], and sex hormone-driven breast cancer [60]. Therefore, in theory, RANKL may potentially impact the reproductive system. Recently, a well-designed study demonstrated that RANKL regulates male reproductive function. By inhibiting RANKL, male fertility and sperm counts increased in both infertile mice and men [61], which shows the endocrine interaction between gonads and bone. In terms of females, a study using oligonucleotides microarrays found overexpressed RANKL in cumulus cells during human oocyte maturation [62]. Moreover, other studies found that RANKL is a candidate gene for age at menarche (AAM) in Caucasian and Chinese women [63,64]. The authors suggested that RANKL may contribute to sexual maturity and puberty via the hypothalamic–pituitary–gonadal (HPG) axis and interactions with hormones. Although no association between RANKL and sex hormones was detected in our cohort, studies specifically designed to assess the potential reproductive impact of RANKL on PCOS patients are warranted in the future.

This study has several limitations. First, as a cross-sectional study, we did not investigate the causal relationships between RANKL and NAFLD incidence. Given the results of this preliminary study, further targeted prospective research on PCOS with NAFLD, especially in non-obese patients, has important significance. Second, we did not measure biochemical markers of bone turnover, which might affect RANKL concentration. Nevertheless, our cohort did not take medications that might affect bone metabolism; thus, the potential impact of these markers was ruled out. Third, we diagnosed NAFLD using ultrasonography. Liver biopsy is the gold standard for diagnosing NAFLD but is not commonly used in clinical practice because of its invasive characteristics. 

## 5. Conclusions

Our study reveals a possible association between RANKL and increased NAFLD risks in PCOS women, regardless of metabolic and reproductive factors. Our findings may provide original insights into the pathogenesis of NAFLD in the PCOS population. The study innovatively explored the association between osteokine RANKL and NAFLD, indicating that the skeletal system might play a role in the energy metabolism of PCOS patients. This finding may also help to identify novel therapeutic targets for NAFLD. More extensive prospective studies and experimental research are necessary to illustrate the exact mechanisms underlying the role of RANKL in NAFLD, especially in non-obese PCOS patients.

## Figures and Tables

**Figure 1 jcm-12-00451-f001:**
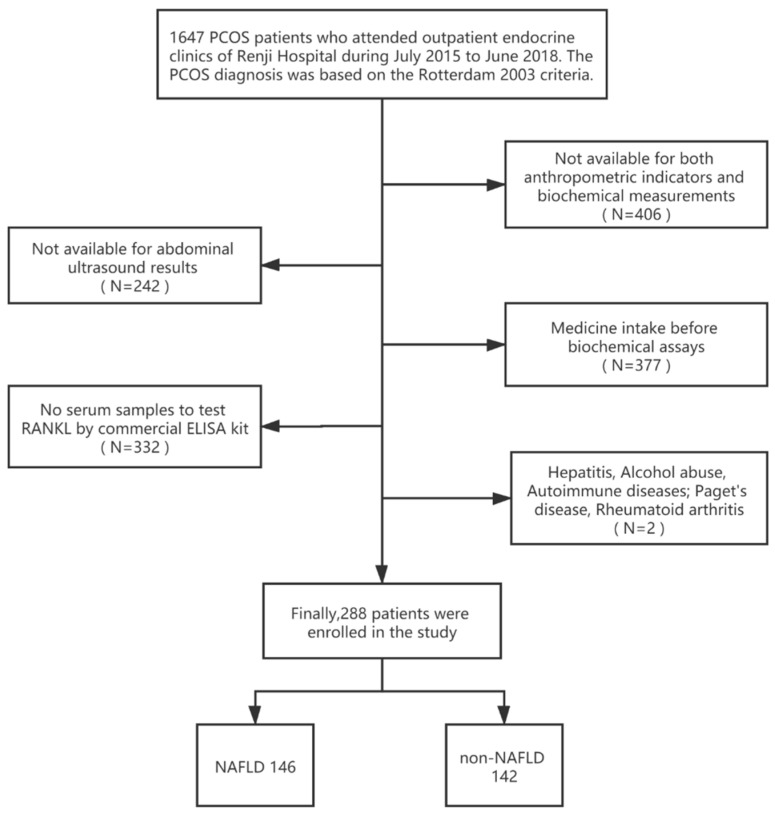
Flow chart of the selection of study participants.

**Figure 2 jcm-12-00451-f002:**
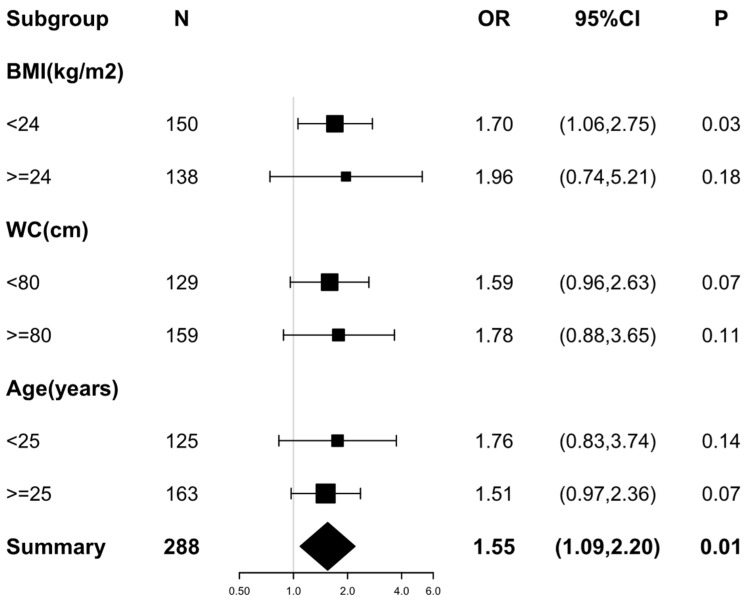
Forest plot of the association between RANKL (per SD) and the risks of non-alcoholic fatty liver disease in different subgroups. Variable adjustments with Age, BMI, WC, hs-CRP, UA, TG, HDL-c, LDL-c, HOMA-IR, LH, FSH, E2, and FAI were performed before the analyses of each subgroup. Odds ratios and 95% confidence intervals are shown as black squares and error bars, respectively. The overall effect is represented by a diamond.

**Table 1 jcm-12-00451-t001:** Biochemical and clinical features of polycystic ovary syndrome patients stratified by the presence of non-alcoholic fatty liver disease (NAFLD).

Variables	Total (*N* = 288)	non-NAFLD (*N* = 142)	NAFLD (*N* = 146)	*p*
Age (years)	25.3 (22–30)	24 (21–28)	27 (23–30.8)	<0.001
BMI (kg/m^2^)	23.7 (21–28.3)	21.2 (19.2–23.1)	28 (24.7–30.1)	<0.001
WC (cm)	82 (73–93)	73.5 (69.3–79)	92 (83.3–95.9)	<0.001
hs-CRP (mg/L)	0.8 (0.3–2.2)	0.4 (0.2–0.9)	1.7 (0.7–3.3)	<0.001
ALT (IU/L)	16 (11–28)	12 (10–16)	25 (16–47.8)	<0.001
UA (μmol/L)	310.5 (266.8–366)	280 (244–313.5)	348.5 (299.5–396.8)	<0.001
TG (mmol/L)	1.1 (0.8–1.6)	0.8 (0.7–1.1)	1.5 (1–2)	<0.001
Tch (mmol/L)	4.6 (4.1–5.2)	4.5 (4–5)	4.8 (4.1–5.6)	0.005
HDL-c (mmol/L)	1.4 (1.1–1.6)	1.5 (1.4–1.8)	1.2 (1–1.3)	<0.001
LDL-c(mmol/L)	2.6 (2.2–3.3)	2.4 (2–3)	2.8 (2.3–3.5)	<0.001
LH (IU/L)	9 (4.8–14.3)	11.1 (5.5–18.2)	8 (4.2–11.5)	<0.001
FSH (IU/L)	6 (4.9–7.4)	6.3 (5.3–7.8)	5.7 (4.5–7)	0.028
PRL (μg/L)	12.2 (9–16.4)	12.8 (8.8–16.8)	11.8 (9.2–16)	0.136
E2 (pmol/L)	185.5 (131.8–247)	194.5 (128.8–248.6)	176.5 (132.3–245.8)	0.462
TT (nmol/L)	2.3 (1.8–2.9)	2.4 (1.9–3)	2.3 (1.6–2.8)	0.231
SHBG (nmol/L)	28.9 (18.1–42.6)	39.8 (28.6–48.6)	19.5 (12.3–29)	<0.001
FAI	8.4 (5.4–13.6)	6.7 (4.7–9.4)	12.1 (7.3–18.5)	<0.001
DHEAs (ng/L)	241.9 (175.5–298)	238.5 (176.5–300.8)	242.0 (172.5–298)	0.782
A2 (μg/L)	3.8 (2.8–4.7)	3.9 (2.8–4.9)	3.7 (2.8–4.5)	0.109
FBG (mmol/L)	4.7 (4.4–5)	4.6 (4.3–4.9)	4.7 (4.4–5.2)	<0.001
FINS (mIU/L)	8.3 (5–13.4)	5.6 (4–8.5)	11.8 (7.9–18.7)	<0.001
HOMA-IR	1.8 (1–2.8)	1.2 (0.8–1.8)	2.6 (1.7–4)	<0.001
RANKL (pg/mL^)^	35.9 (18.3–95.3)	34.0 (19.2–76.2)	40.3 (17.7–134.2)	0.337

Data are expressed as median (interquartile range). *p*-value for non-NAFLD PCOS vs. NAFLD PCOS. Abbreviations: A2, androstenedione; ALT, alanine aminotransferase; BMI, body mass index; DHEAs, dehydroepiandrosterone sulfate; E2, estradiol; FAI, free androgen index; FBG, fasting blood glucose; FINS, fasting serum insulin; FSH, follicle-stimulating hormone; HDL-c, high-density lipoprotein cholesterol; HOMA-IR, homeostasis model assessment of insulin resistance; hs-CRP, high-sensitivity C-reactive protein; LDL-c, low-density lipoprotein cholesterol; LH, luteinizing hormone; PRL, prolactin; RANKL, receptor activator, of nuclear factor-k B liagand; SHBG, sex hormone-binding protein; Tch, total cholesterol; TG, triglyceride; TT, total testosterone; UA, uric acid; WC, waist circumference

**Table 2 jcm-12-00451-t002:** Associations of RANKL levels with clinical variables of PCOS patients.

	RANKL Tertile			*p* for Trend
	Low (≤22.48 pg/mL)	Middle (22.49–59.62 pg/mL)	High (≥59.62 pg/mL)	
Age (years)	27 (23–30)	25 (22–29)	25 (20.8–29)	0.051
BMI (kg/m^2^)	23.8 (21–28)	23.1 (20.4–29.5)	23.8 (21.4–28.3)	0.788
WC (cm)	83 (74.8–92.7)	81.5 (72.3–93)	81 (73–93)	0.484
hs-CRP (mg/L)	0.7 (0.3–1.6)	0.7 (0.2–2.1)	1.1 (0.5–2.3)	**0.023**
ALT (IU/L)	17.5 (11–30.3)	15 (11–20.5)	17.5 (11.8–27.3)	0.559
UA (μmol/L)	321.5 (268.8–372)	301 (264.3–354)	312.5 (263.3–364.5)	0.555
TG (mmol/L)	1.1 (0.8–1.6)	1 (0.7–1.5)	1.2 (0.8–1.7)	0.980
Tch (mmol/L)	4.5 (4–5.1)	4.6 (4.1–5.2)	4.8 (4.1–5.7)	**0.026**
HDL-c (mmol/L)	1.3 (1.1–1.5)	1.4 (1.1–1.7)	1.3 (1.1–1.6)	0.345
LDL-c(mmol/L)	2.5 (2.2–3.1)	2.5 (2.1–3.2)	2.8 (2.2–3.6)	**0.035**
LH (IU/L)	9.1 (4.4–15.6)	9.1 (5.1–13.6)	8.6 (5.3–14.2)	0.792
FSH (IU/L)	5.9 (4.8–7.3)	6 (5–7.5)	6.1 (4.9–7.3)	0.642
PRL (μg/L)	12.1 (9–16.8)	13 (9.4–17)	11.8 (8.9–15)	0.376
E2 (pmol/L)	170.5 (123.3–238.3)	203 (136.8–257)	185 (127.3–240.3)	0.667
TT (nmol/L)	2.4 (1.9–2.9)	2.5 (1.9–2.9)	2.2 (1.7–2.8)	0.448
SHBG (nmol/L)	28.2 (19.5–41.8)	31.8 (18.2–44.3)	25.1 (15.6–42.3)	0.494
FAI	8.3 (5.8–12.8)	8.4 (5.2–13.7)	8.4 (5.9–14.5)	0.341
DHEAs (ng/L)	247 (179.8–326.5)	225 (150.8–291.3)	242 (191.8–286.8)	0.342
A2 (μg/L)	3.9 (2.9–5)	3.8 (2.9–4.5)	3.5 (2.6–5.1)	0.851
FBG (mmol/L)	4.7 (4.4–5)	4.7 (4.3–4.9)	4.7 (4.4–5)	0.451
FINS (mIU/L)	8.4 (5.4–11.6)	7.1 (4.7–13.5)	9.2 (5.1–14.4)	0.191
HOMA-IR	1.8 (1.1–2.6)	1.5 (0.9–2.7)	2 (1.1–3)	0.183

Data are expressed as median (interquartile range). *p* for trend was analyzed by linear regression. Significant *p*-values (*p* < 0.05) are highlighted in bold. Abbreviations: A2, androstenedione; ALT, alanine aminotransferase; BMI, body mass index; DHEAs, dehydroepiandrosterone sulfate; E2, estradiol; FAI, free androgen index; FBG, fasting blood glucose; FINS, fasting serum insulin; FSH, follicle-stimulating hormone; HDL-c, high-density lipoprotein cholesterol; HOMA-IR, homeostasis model assessment of insulin resistance; hs-CRP, high-sensitivity C-reactive protein; LDL-c, low-density lipoprotein cholesterol; LH, luteinizing hormone; PRL, prolactin; RANKL, receptor activator, of nuclear factor-k B liagand; SHBG, sex hormone-binding protein; Tch, total cholesterol; TG, triglyceride; TT, total testosterone; UA, uric acid; WC, waist circumference.

**Table 3 jcm-12-00451-t003:** Independent effect of RANKL (per SD) on NAFLD.

	OR	95% CI	*p*
Model 1	1.411	1.049–1.897	0.023
Model 2	1.469	1.052–2.054	0.025
Model 3	1.545	1.086–2.199	0.015

Model 1: Adjusted for Age, BMI, WC. Model 2: Adjusted for Age, BMI, WC, hs-CRP, UA, TG, HDL-c, LDL-c and HOMA-IR. Model 3: Adjusted for Age, BMI, WC, hs-CRP, UA, TG, HDL-c, LDL-c, HOMA-IR, LH, FSH, E2 and FAI. OR, odds ratio; 95% CI, 95% confidence interval.

**Table 4 jcm-12-00451-t004:** Association between RANKL (per SD) and NAFLD in different subgroups stratified by BMI, WC and Age.

	BMI			
	<24 kg/m^2^ (*n* = 150)		≥24 kg/m^2^ (*n* = 138)	
	OR (95% CI)	*p* value	OR (95% CI)	*p* value
Model 1	1.45 (1.02, 2.06)	**0.04**	2.06 (0.85, 4.94)	0.11
Model 2	1.61 (1.06, 2.45)	**0.02**	1.88 (0.76, 4.68)	0.17
Model 3	1.70 (1.06, 2.75)	**0.03**	1.96 (0.74, 5.21)	0.18
	**WC**			
	**<80 cm (*n* = 129)**		**≥80 cm (*n* = 159)**	
	OR (95% CI)	*p* value	OR (95% CI)	*p* value
Model 1	1.34 (0.92, 1.96)	0.13	1.71 (0.95, 3.10)	0.08
Model 2	1.46 (0.93, 2.29)	0.10	1.82 (0.91, 3.63)	0.09
Model 3	1.59 (0.96, 2.63)	0.07	1.78 (0.88, 3.65)	0.11
	**Age**			
	**<25 years (*n* = 125)**		**≥25 years (*n* = 163)**	
	OR (95% CI)	*p* value	OR (95% CI)	*p* value
Model 1	1.58 (0.88, 2.86)	0.13	1.32 (0.92, 1.88)	0.13
Model 2	1.56 (0.81, 3.00)	0.18	1.38 (0.90, 2.11)	0.14
Model 3	1.76 (0.83, 3.74)	0.14	1.51 (0.97, 2.36)	0.07

Model 1: Adjusted for Age, BMI, WC. Model 2: Adjusted for Age, BMI, WC, hs-CRP, UA, TG, HDL-c, LDL-c and HOMA-IR. Model 3: Adjusted for Age, BMI, WC, hs-CRP, UA, TG, HDL-c, LDL-c, HOMA-IR, LH, FSH, E2 and FAI. Significant *p*-values (*p* < 0.05) are highlighted in bold.

## Data Availability

The data supporting this study’s findings are available from the corresponding author upon reasonable request.

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
