# Peer review of "RANKL Is Independently Associated with Increased Risks of Non-Alcoholic Fatty Liver Disease in Chinese Women with PCOS: A Cross-Sectional Study"

_jcm, 2023, doi:10.3390/jcm12020451_

Round 1

Reviewer 1 Report

In the current study, the authors reported a possible association between RANKL and increased NAFLD risks in PCOS women providing new insights into the pathogenesis of NAFLD in the PCOS population. In a clinical perspectibve the present findings would help to identify novel therapeutic targets for NAFLD. The manuscript is clear and well written.

In a clinical setting, the authors should further discuss the reproductive counselling of these patients in order to show a putative impact, if any, on female reproductive potential.

Author Response

Many thanks for your effort in reviewing our manuscript. We really appreciate your comments and suggestion. We also further polished the language through the editing service company following the suggestions from other reviewers to improve the manuscript's readability. Please find the point-to-point response below and the revised manuscript in the attachment.

Point 1:

In a clinical setting, the authors should further discuss the reproductive counselling of these patients in order to show a putative impact, if any, on female reproductive potential.

Response 1:

We have added further discussion about the potential effects of RANKL on female reproductive function as follows “In the skeletal system, estrogen is one of the important regulators of RANKL, whereas few studies have examined the potential effect of RANKL on female fertility. However, as a member of TNF superfamily, RANKL plays multiple roles through activation of the NF-kB signaling pathway and has been confirmed to be a vital player in immunity[19], mammary gland development during pregnancy[59], and sex hormone-driven breast cancer[60]. Therefore, in theory, RANKL may potentially impact the reproductive system. Recently, a well-designed study demonstrated that RANKL regulates male reproductive function. By inhibiting RANKL, male fertility and sperm counts increased in both infertile mice and men[61], which shows the endocrine interaction between gonads and bone. In terms of females, a study using oligonucleotides microarrays found overexpressed RANKL in cumulus cells during human oocyte maturation[62]. Moreover, other studies found that RANKL is a candidate gene for age at menarche (AAM) in Caucasian and Chinese women[63,64]. The authors suggested that RANKL may contribute to sexual maturity and puberty via the hypothalamic–pituitary–gonadal (HPG) axis and interaction with hormones. Although no association between RANKL and sex hormones was detected in our cohort, studies specifically designed to assess the potential reproductive impact of RANKL on PCOS patients are warranted in the future.” (Line 344-360, page 11, revised manuscript).

Reviewer 2 Report

In this manuscript by Lu et al., a cross-sectional observational study is applied to recently diagnosed PCOS patients that either exhibit or do not exhibit clinical signatures of NAFLD. The goal of the study was to determine whether serum levels of RANKL, which is implicated in hepatic insulin resistance, is associated with NAFLD in PCOS patients. The authors found that total serum RANKL levels are associated with increased risk of NAFLD, especially for non-obese PCOS patients, which is a bit surprising. Overall, this is a useful manuscript. The introduction and discussion are nicely organized with appropriate citations, and the data seem to be appropriately interpreted. I have a few comments/suggestions:

Line 126: Suggest to remove the words “in a refrigerator”. It is technically a low-temperature freezer, but it is sufficient to just report the storage temperature.

Tables 1 and 2: Many variables (Age, BMI, WC, ALT, UA, E2, DHEA, etc.) end in round numbers (e.g., xx.00 or xx.50), which looks a bit odd. Rather than having significant figures to two decimal places for all variables, could the authors just report the values with the number of significant figures in the assay or measurement?

Author Response

Many thanks for your effort in reviewing our manuscript. We really appreciate your comments and suggestions. We have carefully gone through all your comments and tried our best to improve the manuscript. We also further polished the language through the editing service company following the suggestions from other reviewers to improve the manuscript's readability. Please find the point-to-point response below and the revised manuscript in the attachment.

Point 1:

Line 126: Suggest to remove the words “in a refrigerator”. It is technically a low-temperature freezer, but it is sufficient to just report the storage temperature.

Response 1:

Thank you for your correction. We agree with you and remove the words “in a refrigerator”.

Point 2

Tables 1 and 2: Many variables (Age, BMI, WC, ALT, UA, E2, DHEA, etc.) end in round numbers (e.g., xx.00 or xx.50), which looks a bit odd. Rather than having significant figures to two decimal places for all variables, could the authors just report the values with the number of significant figures in the assay or measurement?

Response 2:

Thanks for this helpful suggestion. We have revised the results in tables 1 and 2 by only reporting the values with the number of significant figures.

Reviewer 3 Report

This manuscript by Lu end colleagues is a well-designed cross sectional study. The authors contribute to the literature with one of the first articles that can reveal a possible association between the receptor activator of NF-κB ligand (RANKL) and non-alcoholic fatty liver disease (NAFLD). Furthermore, it can serve as a starting point for several more studies which can identify novel therapeutic targets for NAFLD. Exclusion criteria are a strength of this article as all participants with conditions that can change results are accurately not included. Moreover, the authors are very clear in describing the limitations of this transversal study. I would recommend that this study be extensively reviewed in English to improve the clarity of some important concepts.

Author Response

Many thanks for your effort in reviewing our manuscript. We really appreciate your comments and suggestions. We further polished the language through the editing service company to improve the manuscript's readability. Please find the revised manuscript in the attachment.
